# Non-relativistic torque and Edelstein effect in non-collinear magnets

Rafael González-Hernández[1] ✉, Philipp Ritzinger [2], Karel Výborný[2], Jakub Železný[2] ✉ & Aurélien Manchon [3] ✉

The Edelstein effect is the origin of the spin-orbit torque: a current-induced torque that is used for the electrical control of ferromagnetic and anti-ferromagnetic materials. This effect originates from the relativistic spin-orbit coupling, which necessitates utilizing materials with heavy elements. Here, we show that in magnetic materials with non-collinear magnetic order, the Edelstein effect and, consequently, a current-induced torque can exist even in the absence of the spin-orbit coupling. Using group symmetry analysis, model calculations, and realistic simulations on selected compounds, we identify large classes of non-collinear magnet candidates and demonstrate that the current-driven torque is of similar magnitude as the celebrated spin-orbit torque in conventional transition metal structures. We also show that this torque can exist in an insulating material, which could allow for highly efficient electrical control of magnetic order.

In materials and heterostructures with spin-orbit coupling, the interconnection between the spin and momentum degrees of freedom of the electronic Bloch states underscore a rich landscape of microscopic "spin-orbitronics" phenomena, such as anomalous Hall effect[1] and anisotropic magnetoresistance[2], spin Hall effect[3], Dzyaloshinskii-Moriya interaction or spin-orbit torques[4,5]. To maximize these effects, materials displaying reasonably large spin-orbit coupling are necessary, which implies using metals with large atomic numbers Z, such as Pt, W, Bi, etc. Some of these elements are however scarce, expensive, and environmentally unfriendly. In addition, arbitrarily large spin-orbit coupling does not necessarily lead to arbitrarily large spin-orbitronics phenomena[6,7] because of the competition with crystal field and exchange.

Contrary to a common conception though, spin-orbit coupling is not a mandatory ingredient to obtain spin-momentum locking. In fact, as noticed by Pekar and Rashba in the mid-sixties[8], electronics states in materials with a spatially inhomogeneous magnetization display a spin texture in momentum space that share similarities with the one obtained through spin-orbit coupling. In other words, non-collinear magnetism mimics spin-orbit coupling to some extent and can support a number of phenomena that are well known in spin-orbit coupled

materials such as electric-dipole spin resonance[8,9], topological Hall effect[10–12], spin Hall effect[13], and magnetic spin Hall effect[14,15], the latter being specific to magnetic materials. It is therefore natural to wonder whether another hallmark of spin-orbit coupled materials, the Edelstein effect[16–18] (also called the Rashba-Edelstein effect, inverse spin-galvanic effect, or the magneto-electric effect), and its associated spin-orbit torque can also be achieved in spin-orbit free non-collinear magnets.

The Edelstein effect refers to the generation of nonequilibrium spin density by an applied electric field in non-centrosymmetric semiconducting or metallic materials and heterostructures with spin-orbit coupling. The magnitude of the nonequilibrium spin density is governed by the competition between the spin-orbit coupling energy and the crystal field energy associated with inversion symmetry breaking. In magnetic materials, the spin-momentum locking is governed by the magnetic exchange between local and itinerant electrons, rather than by the atomic spin-orbit coupling, suggesting that a large Edelstein effect can be obtained in non-centrosymmetric magnetic materials. A possible advantage of such a mechanism is that it does not require the presence of heavy elements, and it could exist even in materials with negligible spin-orbit coupling such as organic magnetic

[1]Grupó de Investigación en Física Aplicada, Departamento de Física, Universidad del Norte, Barranquilla, Colombia. [2]Institute of Physics, Czech Academy of Sciences, Cukrovarnická 10, 162 00 Praha 6, Czech Republic. [3]Aix-Marseille Université, CNRS, CINaM, Marseille, France. ✉ e-mail: rhernandezj@uninorte.edu.co; zeleznyj@fzu.cz; aurelien.manchon@univ-amu.fr

materials. In addition, since the magnitude of the Edelstein effect is directly related to the magnetic configuration of the material, it should be highly tunable using an external magnetic field.

Substantial attention has been paid lately to the non-relativistic momentum-space spin texture and splitting in antiferromagnets. Non-collinear antiferromagnets display a spin texture in momentum space that results in the so-called magnetic spin Hall effect, i.e., a transverse spin current whose polarization is governed by the magnetic configuration[14,15]. Collinear antiferromagnets whose sublattices are connected by a rotation symmetry, recently classified as "altermagnets"[19], display momentum-symmetric spin-splitting[20–31] that also supports spin currents[28,32]. These spin currents, which do not generate torque in the bulk material, can exert a self-torque when the antiferromagnet is implemented in a junction[33]. Whereas all these studies have focused on centrosymmetric antiferromagnets displaying momentum-symmetric spin texture and splitting, it has been recently shown that non-centrosymmetric antiferromagnets naturally display momentum-antisymmetric spin texture and splitting[34–36]. This non-relativistic antisymmetric spin-splitting gives rise to a global non-relativistic Edelstein effect at the level of the antiferromagnetic unit cell.

In the present work, we demonstrate that wide classes of anti-ferromagnets lacking a center of inversion, either locally or globally, can support the current-driven Edelstein effect in the absence of spin-orbit coupling. This effect does not necessarily necessitate momentum-antisymmetric spin-splitting and, as such, it is not limited to non-centrosymmetric antiferromagnets but concerns a much broader class of systems including heterostructures and centrosymmetric antiferromagnets. We establish general symmetry principles to discover new materials, propose selected promising candidates, demonstrate and quantify the effect in specific materials, and extend the idea to the case of magnetic multilayers. We implemented an algorithm for determining the symmetry of the non-relativistic Edelstein effect as well as other non-relativistic phenomena and we released it within an open-source code. Remarkably, we show that the non-relativistic Edelstein effect can also be present in insulating materials. This could allow for controlling the magnetic order by a voltage in the absence of any Ohmic conduction, resulting in a much higher efficiency than the conventional current-induced torques.

## Results

### Conditions for an antisymmetric spin texture

In non-magnetic materials lacking inversion symmetry, the relativistic Edelstein effect is associated with an antisymmetric spin texture in the reciprocal space[37]. These spin textures arise from the spin-momentum locking imposed by the spin-orbit coupling and are characterized by a spin direction that varies in momentum space. In the absence of spin-orbit coupling and in the presence of non-collinear magnetism, one expects non-relativistic analogs of the antisymmetric spin textures. Therefore, before addressing the non-relativistic Edelstein effect and its associated torque, we first consider the conditions of the emergence of such antisymmetric spin textures. Recently, spin textures in the absence of spin-orbit coupling have been studied in non-collinear[14,29,34] as well as in collinear magnetic materials[20–31]. In collinear systems, however, the direction of spin is fixed and only the magnitude and sign of the spin-splitting varies in momentum space. In addition, most of the non-relativistic spin textures studied so far (with the exception of ref. 34) are typically symmetric in momentum $\mathbf{k}$, $\mathbf{S}_{n\mathbf{k}} = \mathbf{S}_{n-\mathbf{k}}$, $n$ being the band index, which forbids the realization of the non-relativistic Edelstein effect at the level of the magnetic unit cell.

In the absence of relativistic spin-orbit coupling, the spin and orbital degrees of freedom are decoupled, which also means that the spin is not coupled to the lattice. In such a case the symmetry of magnetic systems is described by the so-called spin space groups[38,39]. In addition to crystallographic symmetry operations that form the

magnetic space groups, which describe the relativistic symmetry of magnetic systems, the spin space groups also contain pure spin rotations. Elements of the spin space groups can be written in the form $\{R_s||R|\boldsymbol{\tau}\}$, where $R_s$ denotes the spin rotation, $R$ is a crystallographic point group operation, i.e., a proper or improper rotation, and $\boldsymbol{\tau}$ is a translation. We denote symmetry operations that contain time reversal as $\{R_s||R|\boldsymbol{\tau}\}'$.

In a 3D periodic system, the rules for the existence of spin-splitting are simple to determine. For an arbitrary $\mathbf{k}$-point (that is, a $\mathbf{k}$-point lying away from any high-symmetry lines or planes), the only symmetry operations that can keep the spin invariant are the combined space-inversion and time reversal (the so-called $P\mathcal{T}$ symmetry), a pure spin rotation, translation, or any combination of these symmetry operations. In a $P\mathcal{T}$ symmetric system, the bands with opposite spin must be degenerate, which is known as Kramers degeneracy and holds even in the presence of spin-orbit coupling. If a pure spin rotation is present, the spin of all non-degenerate states must lie along the spin-rotation axis. If more than one spin rotation with different spin axes is present, this cannot be satisfied for non-degenerate states and thus implies a spin degeneracy. This can also be seen from the fact that spin rotations along different axes do not commute. Since translation does not change spin, the same conclusions apply to symmetry operations that contain translation. Thus spin-splitting can exist in all systems, except those that have a $P\mathcal{T}$ symmetry or two spin rotation axes in the point group. In ferromagnetic systems, spin-splitting can exist anywhere in the Brillouin zone since no symmetry operations connecting states with opposite spin exist. In spin-split antiferromagnetic materials, there can be specific high-symmetry points where opposite spin must be degenerate. This has been studied systematically for collinear antiferromagnets[20–31,40,41]. Note that the spin-split collinear antiferromagnets have sometimes been referred to as "altermagnets"[19].

In a collinear magnetic system, any spin rotation along the magnetic axis is a symmetry. Thus if there exists another spin rotation around a perpendicular axis, the bands must be degenerate. Such a spin rotation must contain translation (otherwise the system could not be magnetic) and can only be a 180° rotation, which in a collinear system has the same effect on the magnetic order as the time reversal. The existence of such a symmetry thus implies that the system is invariant under a $\mathcal{T}\boldsymbol{\tau}$ (combined time reversal and translation) symmetry. Collinear magnetic systems can thus be separated into three types. In systems with $P\mathcal{T}$ symmetry, bands are spin degenerate even with spin-orbit coupling. In systems with $\mathcal{T}\boldsymbol{\tau}$ but broken $P\mathcal{T}$ symmetry, spin-splitting occurs only when the spin-orbit coupling is present. Finally, in systems with broken $P\mathcal{T}$ and $\mathcal{T}\boldsymbol{\tau}$ symmetries, a non-relativistic spin-splitting can be present. Such systems include both ferromagnets as well as antiferromagnets (we use the term antiferromagnet here to refer generally to all magnetically ordered systems with negligible net magnetization). We note that this separation does not hold for non-collinear magnets since, in those, the time reversal does not have the same effect as a 180° spin rotation, and spin rotations with different angles can also occur. Consequently, there can be non-collinear antiferromagnets with $\mathcal{T}\boldsymbol{\tau}$ symmetry that exhibit non-relativistic spin-splitting. This was recently studied in ref. 36.

The existence of antisymmetric spin textures is governed by symmetries that transform $\mathbf{k} \to -\mathbf{k}$. This involves, in particular, the inversion symmetry, which implies $\mathbf{S}_{n\mathbf{k}} = \mathbf{S}_{n-\mathbf{k}}$. In systems with inversion symmetry, any spin texture thus must be symmetric. In a coplanar system a combined spin rotation and time-reversal operation $\{R_s(\hat{\mathbf{n}}_\perp, 180°)||E||E\}'$ is a symmetry. Here $R_s(\hat{\mathbf{n}}_\perp, 180°)$ denotes a spin rotation by 180° around the direction perpendicular to the magnetic plane $\hat{\mathbf{n}}_\perp$. As a consequence, it must hold that $\mathbf{S}_{n\mathbf{k}}^{||} = \mathbf{S}_{n-\mathbf{k}}^{||}$ and $\mathbf{S}_{n\mathbf{k}}^{\perp} = -\mathbf{S}_{n-\mathbf{k}}^{\perp}$, where $\mathbf{S}^{||}$ and $\mathbf{S}^{\perp}$ denote the components of spin parallel and perpendicular to the plane, respectively. In a coplanar system, the only antisymmetric component is thus perpendicular to the magnetic plane, as in the case studied by Hayami et al.[34]. We note that even when

all magnetic moments lie within a plane, the electron spins can contain an out-of-plane component. In a collinear magnetic system, this is not possible, however, since, in this case, spin is a good quantum number and all spins must lie along a single axis. There, any non-relativistic spin-splitting must thus be symmetric in momentum.

## Conditions for nonequilibrium spin densities and torques

Let us now turn our attention toward the non-relativistic Edelstein effect. The nonequilibrium properties of materials obtained via the Kubo formula are often parsed into so-called Fermi surface and Fermi sea contributions[42,43], the former being even under $\mathcal{T}$ and the latter being odd. In the context of the Edelstein effect, the $\mathcal{T}$-even Fermi surface contribution is related to the antisymmetric spin texture in momentum space[16,17], whereas the $\mathcal{T}$-odd Fermi sea contribution is related to the Berry curvature in mixed spin-momentum space in the weak scattering limit[44–46]. As a consequence, in spin-orbit coupled non-centrosymmetric magnetic heterostructures, the Fermi surface contribution produces the so-called field-like torque whereas the Fermi sea contribution is responsible for the antidamping-like torque[5]. Notice that the $\mathcal{T}$-odd Fermi sea contribution can also be non-zero in $PT$ symmetric antiferromagnets with Kramers degeneracy. Furthermore, for manipulating the magnetic order in more complex magnetic systems, especially in antiferromagnets, the nonequilibrium spin density one should be concerned with is the local one, projected on the magnetic sublattices, rather than the global one, at the level of the magnetic unit cell[42,47]. The $\mathcal{T}$-even component of the local Edelstein effect can be understood as originating from the antisymmetric spin texture obtained upon projecting on the local atom. Such a "hidden" texture can again exist even in systems with Kramers degeneracy[48]. Consequently, the symmetry conditions that allow for the existence of an Edelstein effect and a torque on the magnetic order are distinct from those for the existence of antisymmetric spin textures.

The symmetry of the non-relativistic global and local Edelstein effects and the resulting torque can be determined in a similar fashion as for the relativistic one, just replacing the magnetic space groups with spin groups. The key symmetry that needs to be broken for the existence of the Edelstein effect is the inversion symmetry. This holds regardless of the presence of spin-orbit coupling. For the global Edelstein effect, the global inversion symmetry must be broken, whereas for the local Edelstein effect, it has to be broken locally (e.g., see ref. 48). This means that for the presence of the Edelstein effect on a given magnetic site, there must be no inversion symmetry operation that would leave this site invariant.

As already mentioned, in magnets the Edelstein effect can be non-zero even without spin-orbit coupling, similar to the spin Hall effect, for example[49]. This applies even to collinear magnets; however, in such a case, the induced spin density must be oriented along the magnetic order and does not lead to a torque (although it could play a role, for example, in the presence of magnons). Consequently, we focus here on non-collinear magnetic systems, seeking the symmetry rules that govern the emergence of $\mathcal{T}$-odd and $\mathcal{T}$-even spin densities, respectively referred to as $S^{odd}$ and $S^{even}$. In the presence of a pure spin rotation $\{R_s(\hat{\mathbf{n}}, \theta) || E || \tau\}$, where $\tau$ could also be zero, the global Edelstein effect must obey $S || \mathbf{n}$. In the presence of spin rotation coupled with time reversal $\{R_s(\hat{\mathbf{n}}, \theta) || E || \tau\}'$ it obeys $S^{even} || \mathbf{n}$ and $S^{odd} \perp \mathbf{n}$. The same holds for the local Edelstein effect as long as the site is invariant under $\tau$. Consequently, in coplanar systems, $S^{even}$ must be oriented perpendicular to the magnetic plane and $S^{odd}$ must lie within the plane for both the global and the local Edelstein effects.

To determine the full symmetry of the non-relativistic Edelstein effect, it is necessary to consider all the symmetry operations of the spin group. We have implemented an algorithm for determining all spin group symmetry operations of a given magnetic system within the freely available open-source *Symmetr* code[50]. The process of determining the non-relativistic symmetry is described in detail

in Supplementary materials. We have utilized this program to explore the symmetry of non-collinear materials from the MAGNDATA database of magnetic materials. We have analyzed the symmetry of 484 non-collinear magnetic materials and have found that the global Edelstein effect is allowed in 160 of these materials, whereas the local Edelstein effect on a magnetic sublattice is allowed in 355 compounds. The full list is given in the Supplementary materials. As also described in the Supplementary materials, the Symmetr code allows one to directly obtain the non-relativistic symmetry of the Edelstein effect (as well as other phenomena) for materials from the MAGNDATA.

Among the noticeable materials whose crystal structure admits both a global and local (sublattice) torque, we identified ferroelectric antiferromagnets such as orthorhombic $DyFeO_3$, hexagonal $HoMnO_3$, $YbMnO_3$, and $LuFeO_3$, as well as metallic antiferromagnets such as $\alpha$-Mn, $Tb_3Ge_5$ and $Tb_5Ge_4$. Interestingly, the centrosymmetric metallic antiferromagnets $Mn_5Si_3$, $Mn_3(Sn, Ge, As)$, and $Mn_3CuN$ do not display a global torque but do admit a local torque on the individual magnetic sublattices. These torques are expected to induce magnetic excitations and potentially magnetic order reversal. In the following, we explicitly compute the global and local Edelstein effects in both $LuFeO_3$ and $Mn_3Sn$ as an illustration of both cases.

## Non-relativistic Edelstein effect in non-collinear antiferromagnets

To calculate the Edelstein effect and torque we use the Kubo formula within the constant relaxation time approximation. We only consider an Edelstein effect linear in an electric field: $\delta S_i = \chi_{ij} E_j$, where $\delta S_i$ is the induced spin, $E_j$ is the electric field, and $\chi_{ij}$ is a response tensor. The $\mathcal{T}$-even and $\mathcal{T}$-odd components are computed using the Kubo formula derived in refs. 45,51,

$$\chi_{ij}^{even} = -\frac{e\hbar}{\pi} \sum_{\mathbf{k},m,n} \frac{\mathrm{Re}\left[\langle\psi_{\mathbf{k}n}|\hat{S}_i|\psi_{\mathbf{k}m}\rangle\langle\psi_{\mathbf{k}m}|\hat{v}_j|\psi_{\mathbf{k}n}\rangle\right]\Gamma^2}{\left((\varepsilon_F - \varepsilon_{\mathbf{k}n})^2 + \Gamma^2\right)\left((\varepsilon_F - \varepsilon_{\mathbf{k}m})^2 + \Gamma^2\right)}, \quad (1)$$

$$\chi_{ij}^{odd} = 2e\hbar \sum_{\mathbf{k},n\neq m}^{\substack{n\ occ.\\m\ unocc.}} \mathrm{Im}\left[\langle\psi_{n\mathbf{k}}|\hat{S}_i|\psi_{m\mathbf{k}}\rangle\langle\psi_{m\mathbf{k}}|\hat{v}_j|\psi_{n\mathbf{k}}\rangle\right]$$
$$\times \frac{\Gamma^2 - (\varepsilon_{\mathbf{k}n} - \varepsilon_{\mathbf{k}m})^2}{\left[(\varepsilon_{\mathbf{k}n} - \varepsilon_{\mathbf{k}m})^2 + \Gamma^2\right]^2}. \quad (2)$$

Here $\psi_{\mathbf{k}n}$ is the Bloch function of band $n$, $\mathbf{k}$ is the Bloch wave vector, $\varepsilon_{\mathbf{k}n}$ is the band energy, $\varepsilon_F$ is the Fermi energy, $\hat{v}_j$ is the velocity operator, $e > 0$ is the elementary charge, $\hat{S}_i$ is the spin operator, and $\Gamma$ is a parameter that describes the strength of disorder, which is related to the relaxation time $\tau = \hbar/2\Gamma$. This parameter is usually chosen to match the conductivity computed numerically with the experimental value. To calculate the local Edelstein effect on a given sublattice, a projection of the spin operator on the sublattice is used instead.

In the limit $\Gamma \to 0$, Eq. (1) goes to the semiclassical Boltzmann constant relaxation formula, which scales as $1/\Gamma$ whereas Eq. (2) goes to the so-called intrinsic formula, which is $\Gamma$ independent and can be understood in terms of Berry curvature in mixed spin-momentum space[46]. Equations (1) and (2) are sometimes referred to as "intraband" and "interband" contributions, respectively.

## A non-coplanar 3Q antiferromagnet

An example of a non-relativistic Edelstein effect in a non-collinear coplanar antiferromagnet was recently given by Hayami et al.[34]. In this case, the coplanarity of the magnetic texture imposes the current-driven spin density to be oriented perpendicular to the magnetic plane. Here, we adopt a triangular antiferromagnet with a 3Q spin texture, as displayed in Fig. 1a. This magnetic texture can be stabilized in the presence of 4-spin interaction[52,53] and hosts quantum anomalous Hall effect[10–12]. The 3Q texture is also commonly observed in

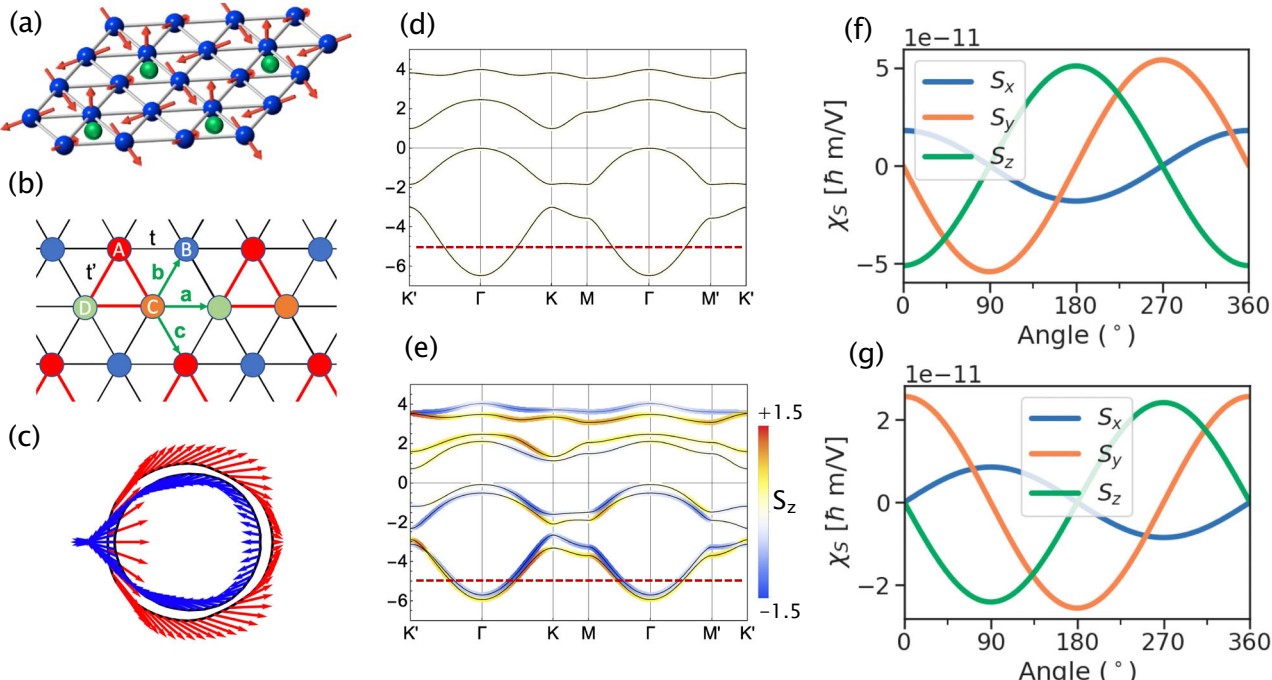

**Fig. 1 | Global torque in the 3Q triangular antiferromagnet. a** Sketch of the triangular lattice with 3Q non-coplanar configuration of the magnetic moments. The blue atoms are magnetic and the green atoms break the planar inversion symmetry. **b** Top view of the triangular lattice. The hopping parameters of the black and red bounds are $t$ and $t'$, respectively. **c** In-plane spin texture in momentum space at energy $\varepsilon = -5$ eV, corresponding to the red dashed line in panel (**e**).

**d, e** Band structure for $t' = t$ and $t' = t/2$, respectively. In the absence of inversion symmetry breaking, $t' = t$, the degenerate bands display compensating spin texture in momentum space. When $t' \neq t$, the band degeneracy is lifted and the spin textures no longer compensate. The color scale indicates the value of $S_z$. **f, g** $\mathcal{T}$-even (**f**) $\mathcal{T}$-odd (**g**) contributions for $t' = t/2$ and $\varepsilon = -4$ eV when rotating the electric field direction in the $(x,y)$ plane. We set $t = 1$ eV, $\Gamma = 0.1$ eV and the exchange is $J = -2$ eV.

three-dimensional materials such as $\gamma$-FeMn[54] and pyrochlores[55]. We use a simple tight-binding model with a 3Q spin texture to illustrate the physical properties of such systems. The model is given by

$$H = - \sum_{\langle ab \rangle \alpha} t_{ab} c^{\dagger}_{a\alpha} c_{b\alpha} + J \sum_{a\alpha,\beta} (\boldsymbol{\sigma} \cdot \mathbf{m}_a)_{\alpha\beta} c^{\dagger}_{a\alpha} c_{a\beta}. \quad (3)$$

Here, $c^{\dagger}$ and $c$ denote the creation and annihilation operators respectively; $a$, $b$ denote the site index and $\alpha$, $\beta$ the spin index. The first term is the nearest-neighbor hopping term, with $t_{ab}$ representing the hopping magnitude. The second term represents the coupling of the conduction electrons to the on-site magnetic moments. Here $\mathbf{m}_i$ is the magnetic moment direction, $J$ is the exchange parameter and $\boldsymbol{\sigma}$ is the vector of Pauli matrices. We only consider nearest-neighbor hopping. To break the inversion symmetry, we use two different hopping magnitudes, as shown in Fig. 1b. This could be understood as due to the presence of another atom illustrated in Fig. 1a.

The band structures of the 3Q antiferromagnet are given in Fig. 1c, f, without and with inversion symmetry breaking. In the absence of inversion symmetry breaking, the band structure is doubly degenerate. Breaking the inversion symmetry lifts the band degeneracy (Fig. 1f) and results in a spin texture, shown in Fig. 1e. This spin texture contains both symmetric and antisymmetric components, the latter giving rise to the current-driven Edelstein effect and its torque. When the inversion symmetry is broken, we observe a finite Edelstein effect as shown in Fig. 1d, g. Several features are worth noticing. First, because the magnetic texture of the 3Q antiferromagnet spans the 3D space, the current-driven spin density possesses all three components, $S_x$, $S_y$ and $S_z$, which strikingly contrasts with the result of ref. 34. Second, both $\mathcal{T}$-even and $\mathcal{T}$-odd components contribute with similar magnitude. Finally, for the set of parameters adopted in this calculation, i.e., the exchange and hopping energies are of comparable magnitude ($\Delta = 2t = 4t' = 2$ eV), we obtain a nonequilibrium spin

density of about $10^{-11}\,\hbar$m/V. For the sake of comparison, in a two-dimensional Rashba gas, the nonequilibrium spin density is[17] $S^R_{\text{surf}}/eE = (\alpha_R/\hbar^2)(m_0/\pi\Gamma)$. Taking $\Gamma = 0.1$ eV, $m_0$ being the free electron mass, $\alpha_R = 10^{-9}$ eV $\cdot$ m as the typical Rashba strength expected in transition metal heterostructures[56], and $(3\,\text{Å})^2$ as a unit cell area, the Edelstein effect yields $\chi_S \sim 3.6 \times 10^{-11}\,\hbar$m/V, which is in the same range as our calculations for the two-dimensional 3Q system reported in Fig. 1.

**A centrosymmetric antiferromagnet: Mn₃Sn**

In antiferromagnets, and in general in more complex magnetic systems, the magnetic dynamics is not determined by the global Edelstein effect, but rather by the local Edelstein effect on each magnetic sublattice. Consequently, in antiferromagnets, it is the local rather than the global inversion symmetry breaking that is necessary for the existence of the Edelstein effect and the current-induced torque[42]. An example of an antiferromagnet with a global inversion symmetry and a local inversion symmetry breaking is the well-known non-collinear antiferromagnet Mn₃Sn[57,58]. In this material, the global Edelstein effect vanishes but the local Edelstein effect is allowed on each sublattice, even in the absence of spin-orbit coupling.

The crystal and magnetic structure of Mn₃Sn are given in Fig. 2c. Mn₃Sn has six magnetic sublattices, which are composed of three pairs of sites with equivalent moments connected by inversion symmetry. Notice that we neglected the small spin canting of Mn₃Sn induced by spin-orbit coupling as it has a negligible influence on the non-relativistic torque we are concerned about. The inversion partners are denoted by′ in Fig. 2b. Due to the inversion symmetry, the Edelstein effect on the two inversion-connected sites must be opposite. The local Edelstein effect tends to drive the system into a state where the magnetic moments of the inversion-connected sites are not parallel and thus it acts against the exchange. As such, it is unlikely to reverse the magnetic order but it can excite different magnon modes. Leaving the rigorous analysis of the magnetic dynamics to future studies, we

emphasize that the global inversion symmetry can be broken, for example, by an interface. Then the two sites with the same moments are no longer connected by inversion symmetry and consequently can experience the Edelstein effect of the same sign, enabling the electric manipulation of the magnetic order.

We evaluate the Edelstein effect in Mn₃Sn using ab initio calculations with and without spin-orbit coupling (see "Methods" for details of the calculation setup). The result of the calculation for $\Gamma = 0.01$ eV is shown in Fig. 2a. We find substantial $\mathcal{T}$-even and $\mathcal{T}$-odd Edelstein effect on all sublattices. Including the spin-orbit coupling does not change the results substantially, similar to previous calculations of the spin Hall effect in this material[13]. Our calculations agree well with the symmetry analysis shown in the Supplementary materials. Notice that,

again, the magnitude of the current-driven spin density is rather large, corresponding to a Rashba strength of $10^{-9} - 10^{-10}$ eV · m.

We note that current-induced switching of Mn₃Sn has been experimentally observed in Mn₃Sn/non-magnetic metal heterostructures[59–62]. The switching has been attributed to the spin Hall effect from the non-magnetic metal layer and to spin-transfer torque and inter-grain spin-transfer torque, however, it is possible that the non-relativistic Edelstein effect also contributes.

## A non-centrosymmetric antiferromagnet: LuFeO₃

As an example of a real non-collinear antiferromagnet that can exhibit a global non-relativistic Edelstein effect and torques, we consider the hexagonal LuFeO₃, a multiferroic with antiferromagnetic order. In bulk, LuFeO₃ is typically orthorhombic. However, the hexagonal phase has been stabilized in thin layers[63] and can also be stabilized in the bulk[64]. It has a non-collinear coplanar antiferromagnetic structure with magnetic space group (MSG) #185.201 (P6₃c'm') as presented in Fig. 3a[63,64]. The inversion symmetry is broken in this material by the crystal structure, which suggests the possibility of non-relativistic spin torques. The system has a small net moment of ~0.02$\mu_B$ along the $z$ direction (weak ferromagnetism). This moment is of relativistic origin, and thus, in the absence of spin-orbit coupling, the magnetic structure is perfectly compensated. Apart from the magnetic order, hexagonal LuFeO₃ also exhibits a ferroelectric order that is present below ~1000K and the material has attracted large attention for its multiferroic properties and the possibility of magneto-electric coupling[63–65].

The non-relativistic electronic structure is shown in Fig. 3b. The material is insulating; here we only show the valence bands, which are relevant to our calculations. As can also be seen in Fig. 3b, the bands are spin-split and thus there is also a non-relativistic spin texture, shown in Fig. 3c for two cuts through the Brillouin zone. Due to the coplanarity of the magnetic order, the spin texture is symmetric for the $S_x$ and $S_y$ components and antisymmetric for the $S_z$ component. The $S_z$ component is non-zero but very small.

For the calculation of the Edelstein effect, we move the Fermi level into the valence band to simulate doping. Our symmetry analysis shown in the Supplementary materials shows that the Edelstein effect is allowed in LuFeO₃ even with no spin-orbit coupling. Results of the calculation for $\Gamma = 0.01$ eV with and without spin-orbit coupling are given in Fig. 4. We calculate both the global Edelstein effect and the local one for all Fe sublattices. For brevity though, we only show here

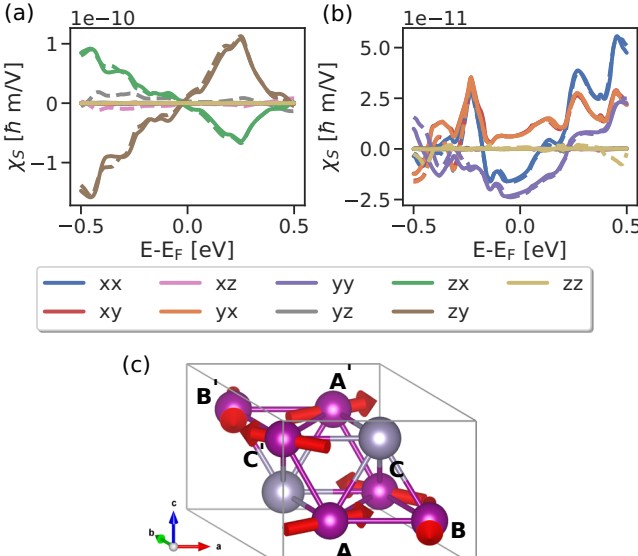

**Fig. 2 | Local torque in centrosymmetric Mn₃Sn. a**, **b** Calculation of the $\mathcal{T}$-even (**a**) and $\mathcal{T}$-odd (**b**) local Edelstein effect in Mn₃Sn with (dashed lines) and without (solid lines) spin-orbit coupling. The individual lines correspond to different tensor components, see Eq. (1)-(2). **c** The crystal structure and magnetic structure of Mn₃Sn. The purple and gray spheres represent the Mn and Sn ions, respectively.

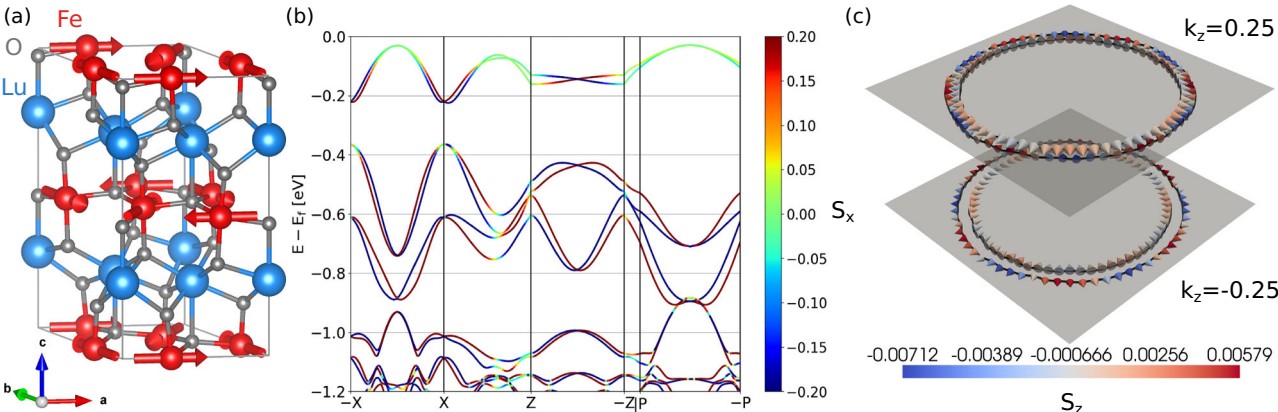

**Fig. 3 | The non-centrosymmetric antiferromagnet LuFeO₃. a** The crystal and magnetic structure of the hexagonal LuFeO₃. **b** LuFeO₃ band structure without spin-orbit coupling. The color denotes the $S_x$ projection. $X$ points represent opposite $k_x$ coordinates, $Z$ points represent opposite $k_z$ coordinates, and $P$ points represent opposite $k_x, k_y, k_z$ coordinates in the Brillouin zone. Asymmetric -*odd* in k- and symmetric -*even*-spin-splitting is labeled in the corresponding k-path. **c** The

spin texture of LuFeO₃ at the Fermi surface for Fermi level 0.45 eV below the top of the valence band. We plot the spin texture for two planes corresponding to $k_z = 0.25$ Å⁻¹ and $k_z = -0.25$ Å⁻¹. The center of the figure lies at the $\Gamma$ point. The arrows represent the spin and we use the color to highlight the $z$-component of the spin since it would be hard to distinguish otherwise.

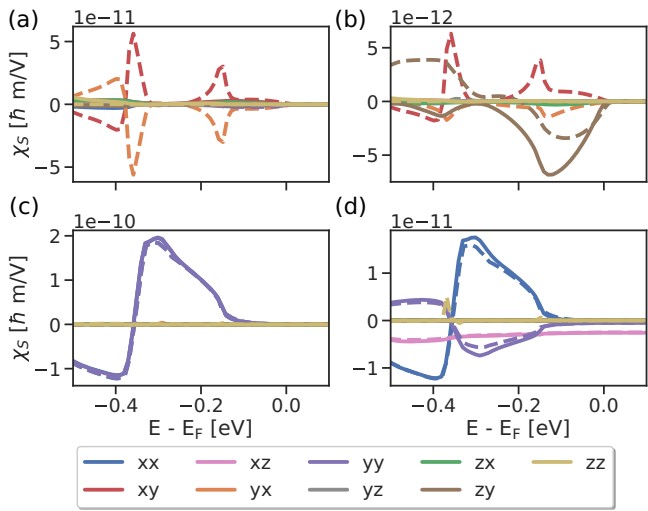

**Fig. 4 | Edelstein effect in LuFeO₃ computed with (dashed lines) and without spin-orbit coupling (solid lines).** Here zero energy corresponds to the top of the valence band. **a** $\mathcal{T}$-even component of the total Edelstein effect. **b** $\mathcal{T}$-even component of the local Edelstein effect. **c** $\mathcal{T}$-odd component of the total Edelstein effect. **d** $\mathcal{T}$-odd component of the local Edelstein effect.

the result for the global effect and for one sublattice. The full results are shown in the Supplementary materials.

Our calculations reveal a large non-relativistic global and local Edelstein effect in good agreement with the symmetry analysis. Without spin-orbit coupling for the global effect, only the $\mathcal{T}$-odd component is allowed (Fig. 4b). With spin-orbit coupling, even the global $\mathcal{T}$-even component appears (Fig. 4a). We find that the effect of spin-orbit coupling is quite small for the $\mathcal{T}$-odd component, but fairly large for the $\mathcal{T}$-even component. For the local effect (Fig. 4c, d), both $\mathcal{T}$-even and $\mathcal{T}$-odd components are allowed.

An important remark is in order. The $\mathcal{T}$-even component has to vanish within the gap since it is a Fermi surface property (see Eq. (1)). The global $\mathcal{T}$-odd component vanishes within the gap as well (Fig. 4b). However, we find that the local $\mathcal{T}$-odd components on the Fe atoms are non-zero within the gap, as shown in Fig. 4d and in the Supplementary materials. Only the *xz* and *yz* components are non-zero within the gap, reaching a constant value. Such a result is intriguing as within the gap, there is no Ohmic conduction, and thus heat dissipation is absent. This could consequently allow for electric field control of magnetic order in the absence of Ohmic dissipation. The existence of spin-orbit torque in an insulator was previously studied in topological materials[46,66,67]. Since the $\mathcal{T}$-odd Edelstein effect is related to the Berry curvature in spin-momentum space, it arises from matrix elements between occupied and unoccupied states which do not necessarily vanish in the gap. Our results are similar, except that in the case of LuFeO₃, the origin of the torque is non-relativistic, due to the coexistence of the non-collinear magnetic order with inversion symmetry breaking. We point out that the torque is not quantized, contrary to the quantized magneto-electric effect in topological insulators[66,68], and therefore unlikely to be of topological origin. We are also not aware of any topological properties of LuFeO₃. The $\mathcal{T}$-odd torque is governed by the Berry curvature in mixed spin-momentum space and involves electrically driven interband transitions, resulting in a finite (but not quantized) value in the gap.

We note that in metals the torques induced by an electric field are accompanied by an electric current and thus often referred to as "current-induced" torques. However, even in metals, the torques are, in fact, due to the electric field rather than to the current flow, although the torque cannot exist without Ohmic conduction. In non-centrosymmetric insulating magnets though, Ohmic conduction is

suppressed while the electrically driven torque remains sizable, as demonstrated in LuFeO₃. This opens promising perspectives for the dissipation-free electrical control of magnetization.

## Non-centrosymmetric heterostructures

The examples we have discussed so far all have inversion symmetry (globally or locally) broken in the bulk of their crystal structure. Such a constraint, however, severely restricts the Edelstein effect to the materials listed in the Supplemental materials. For this reason, we propose to exploit the broken inversion symmetry taking place at the interface between the non-collinear antiferromagnet and an adjacent metal. Such heterostructures are commonly utilized for spin-orbit torque, where the ferro- or antiferromagnet is typically interfaced with a heavy element metal such as platinum[5]. This simple but instrumental configuration allows for observing the spin-orbit torque in a wide variety of systems and enables interfacial engineering of the spin-orbit torque properties.

The same concept can be applied to the non-relativistic Edelstein effect. When a non-collinear magnetic material with inversion symmetry is interfaced with a different material, the broken inversion symmetry can result in a non-relativistic Edelstein effect, which in turn generates a torque on the magnetic order. We illustrate this concept using the example of the well-known non-collinear antiferromagnet Mn₃Ir, whose crystal and magnetic structures are displayed in Fig. 5a. In this material, each magnetic site is an inversion center, and thus no Edelstein effect is allowed in the bulk. To break the inversion symmetry, we consider a thin layer of Mn₃Ir interfaced with a thin layer of a non-magnetic material. When Mn₃Ir is grown along the [001] direction, the non-relativistic Edelstein effect in such a heterostructure is only allowed for an electric field along the [001] direction. In such a case no electric current can flow, however. Thus we instead consider Mn₃Ir grown along the [111] direction. For this orientation, the symmetry is lowered, and the Edelstein effect is allowed for an electric field oriented along the interface. We consider a structure composed of 12 atomic layers of the Mn₃Ir, as shown in Fig. 5b. The individual atomic layers are shown in Fig. 5c.

We utilize a simple tight-binding model that is not meant to give quantitative predictions but rather to confirm that the effect can exist and illustrate its basic properties. The model is analogous to the one we have used for the 3Q antiferromagnet. It is composed of s-electrons on each site with nearest-neighbor hopping and exchange coupling to the atomic magnetic moments. Similar models have been utilized to demonstrate other properties of the Mn₃Ir and similar antiferromagnets such as the non-relativistic spin currents[13] or the anomalous Hall effect[69]. We do not include any spin-orbit coupling in the model. The Hamiltonian is in Eq. (3), where we consider nearest-neighbor hopping $t = 1$ eV and magnetic exchange $J = 1.7$ eV on Mn atoms (gray and gold atoms representing non-magnetic layer and Ir atoms, respectively, are not endowed with magnetic moment).

The calculated Edelstein effect is shown in Fig. 5c for each sublattice and atomic layer. For this calculation, we have used $\Gamma = 0.01$ eV and $\varepsilon_F = 0$ eV. Our calculations are fully in agreement with the symmetry analysis, shown in the Supplementary materials. Both $\mathcal{T}$-even and $\mathcal{T}$-odd components are present. We find the largest effect close to the interfaces, although the current-driven spin density remains sizable in the center of the Mn₃Ir layer. A large effect is found both at the interface with the non-magnetic layer and at the top surface, which illustrates that the presence of another layer is, in principle, not necessary.

In Fig. 5, we only give the result of the calculation for the tensor components that correspond to an in-plane electric field. Interestingly, the out-of-plane components do not vanish even though no current can flow in the out-of-plane direction, similar to the case of the LuFeO₃. In this case, however, since the system is metallic, the out-of-plane electric field is screened, and thus the effect is hard to observe in practice.

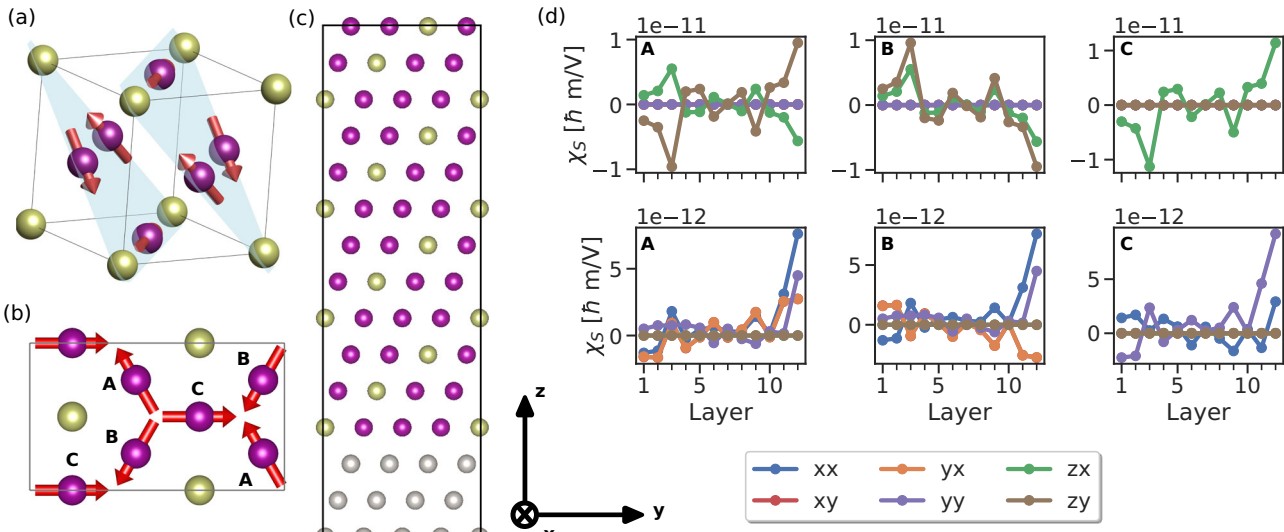

**Fig. 5 | Local torque in an antiferromagnetic heterostructure. a** Mn₃Ir bulk structure, where the purple and golden spheres represent the Mn and Ir ions, respectively. **b** Top-down view of a single atomic layer in the Mn₃Ir [111] bilayer. Here A, B and C denote the three different sublattices. Note that we have constructed the system in such a way that there are 6 Mn atoms in each atomic layer; however, the two atoms for each sublattice are connected by translation and thus there are only three non-equivalent sublattices. **c** Side view of the Mn₃Ir[111] bilayer, where the gray spheres represent the non-magnetic substrate. Here we do not show the magnetic moments. **d** Edelstein effect calculation in the Mn₃Ir [111] for each sublattice as a function of the atomic layer. The top row plots show the $\mathcal{T}$-even component and the bottom plots show the $\mathcal{T}$-odd component.

In addition to the Mn₃Ir bilayer, we also include results of analogous tight-binding calculations for the Mn₃Sn/Ru bilayer in the Supplementary materials. These results show that, in the presence of interfacial inversion symmetry breaking, the compensation of the Edelstein effect on inversion-pair sublattices that naturally occurs in bulk Mn₃Sn is no longer fulfilled. Consequently, a net Edelstein effect emerges, allowing for efficient manipulation of the magnetic order. We have also included preliminary results of first-principles calculations of the Mn₃Sn/Ru bilayer. These calculations, only included for illustration, show that the magnitude of the Edelstein effect in the first-principles calculations is comparable to the results of the tight-binding calculations.

## Discussion

The torque induced by the non-relativistic Edelstein effect shares important similarities with the conventional spin-orbit torque. Both torques are electrically driven self-induced torques that necessitate inversion symmetry breaking, either locally or globally. The key difference though is that the torque due to the non-relativistic Edelstein effect does not originate from spin-orbit coupling but rather from the non-collinear magnetic order. As a consequence, the microscopic origin of these two torques is quite distinct. In the non-relativistic limit, spin is conserved and the torque is directly associated with a spin current: the torque corresponds to spin sources[33] and can be understood as a local transfer of spin angular momentum within the magnetic unit cell.

In the present work, we have computed the non-relativistic Edelstein effect in four different systems, all displaying inversion symmetry breaking, either in the magnetic unit cell or locally. It is quite remarkable that all the examples discussed here display a sizable electrically induced spin density, in spite of the absence of spin-orbit coupling. For the sake of comparison, in our previous calculations of the relativistic Edelstein effect in a collinear antiferromagnet Mn₂Au, we found a magnitude of $\chi_S \sim 4.3 \times 10^{-11}$ ℏm/V for $\Gamma = 0.01$ eV[42], corresponding to a Rashba strength of $10^{-9}$ eV · m, similarly to the magnitude reported in our realistic simulations on LuFeO₃ and Mn₃Sn. Further systematic studies as necessary to determine the conditions for a maximal non-relativistic Edelstein effect.

A central feature of the non-relativistic nature of the torque is its dependence on the magnetic order. In most magnetic systems, the magnetic exchange is much larger than any other magnetic interactions or torques acting on the system. Hence, during the dynamics of the magnetic order, the angles between the individual magnetic moments stay approximately unchanged. Therefore, the dynamics of the magnetic order are described by an overall rotation of all magnetic moments and a small canting. In the non-relativistic limit (ignoring the small canting) the rotated states are connected by a spin rotation and, consequently, the corresponding torques must also be transformed by this spin rotation. Specifically, any torque acting on magnetic moment $\mathbf{M}_i$ can be written as $\mathbf{T}_i = \mathbf{M}_i \times \mathbf{B}_i$, where $\mathbf{B}_i$ is an effective magnetic field. When the magnetic moments are rotated by rotation $R$ then the torque reads $\mathbf{T}_i = R\mathbf{M}_i \times R\mathbf{B}_i$. This is quite distinct from the conventional spin-orbit torque for which the two most important terms are the field-like torque, in which $\mathbf{B}_i$ is independent of $\mathbf{M}$, and the antidamping torque, in which $\mathbf{B}_i \sim \mathbf{M}_i \times \mathbf{p}$, $\mathbf{p}$ being some constant direction[5].

Because of the dependence of the non-relativistic torque on the magnetic order, it may be difficult to realize reversible switching since there can be no magnetic configuration for which the effective field $\mathbf{B}_i$ vanishes. This might not be such a limitation in practice, however, since some spin-orbit coupling is always present, which may enable deterministic switching even in cases where the non-relativistic torque is dominant. Furthermore, deterministic switching could be achieved by using field-assisted switching or precise pulse timing. In the presence of antiferromagnetic domain walls, the non-relativistic torque could provide an additional source of spin current and therefore enhance or quench the domain wall mobility, depending on the wall configuration. In fact, the very dependence of the torque on the magnetic ordering makes the interplay between the flowing electrons and the magnetic order particularly rich and, as such, the non-relativistic torque is well adapted to excite magnetic modes and self-oscillations which, in antiferromagnets, are particularly appealing for THz applications.

## Methods

The DFT calculations use the VASP code[70] and we use the Wannier90 code[71] to construct the Wannier Hamiltonian that serves as an input to

the linear response calculations. For LuFeO3 we use $9 \times 9 \times 3$ k-point mesh and 520 eV cutoff energy. For the Wannierization, we use the $s$ and $d$ orbitals for Fe, $s$ and $p$ orbitals for O and $\varepsilon_F + 3$ eV frozen energy window. For Mn3Sn we use $11 \times 11 \times 11$ k-point mesh and set the cutoff energy to 520 eV. For the Wannierization, we use the $s$, $p$, and $d$ orbitals for the Mn atoms, $s$ and $p$ orbitals for the Sn atoms, and we set the frozen energy window to $\varepsilon_F \pm 2$ eV.

For the linear response calculations, we use the Linres code[72]. This code uses the Wannier or tight-binding Hamiltonian as an input. This Hamiltonian is then Fourier transformed to a dense mesh in the reciprocal space, which is used for evaluating the Kubo formulas as described in ref. 14.

## Data availability
The data that support the findings of this study are available from the corresponding authors upon request.

## Code availability
The open-source codes used in the work are available on the following links: refs. 50,72.

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

## Acknowledgements

R.G.-H. was supported by Universidad Nacional de Colombia with QUIPU code 202010042199 and MinCiencias Conv. 937 Fundamental Research. P.R. and K.V. acknowledge the Grant Agency of the Czech Republic Grant No. 22-21974S. J.Z. acknowledges support from the Dioscuri Program LV23025 funded by MPG and MEYS and TERAFIT-CZ.02.01.01/00/22_008/0004594. A.M. acknowledges support from the Excellence Initiative of Aix-Marseille Université, A*Midex, a French "Investissements d'Avenir" program, as well as from King Abdullah University of Science and Technology (KAUST) under award 2022-CRG10-4660.

## Author contributions

A.M. and J.Z. conceived the idea, R.G.-H., J.Z., and P.R. performed the first-principles and tight-binding calculations, and J.Z. performed the symmetry analyses. All the authors contributed to the discussions of the results and the preparation of the manuscript.

## Competing interests

The authors declare no competing interests.
