## [Peer Review File · Nature Communications]

Reviewers' Comments:

Reviewer #1:

Remarks to the Author:

The authors have discussed the torque and Edelstein effects emerged in noncollinear magnets where the relativistic spin-orbit coupling plays a negligible role. They first discussed symmetry conditions for the presence of spin splitting in particular noncolinear magnetic structure with 3 components, and revealed that the directions of S^{even} and S^{odd} are classified by symmetry operations. Then, they demonstrate the spin splitting and Edelstein effect for some representative materials. They also found that in some class of compounds, the non-relativistic origin of magnetic torque exists in insulators. The contents are worth publishing in some form, but the impact of the manuscript is unclear. The potential weaknesses are: (i) it is hard to extract the authors' new assertions as compared with the previous relevant works, and (ii) the authors seem to estimate the responses quantitatively for realistic heterostructure, but the model parameters are chosen as arbitrary, which means that quantitative conclusions are valid or not in general sense and it is not clear which conclusion is applied widely to similar compounds in general. By using the heterostructures having lower symmetry, it is of course expected that the non-relativistic origin of spin splitting and consequent Edelstein and magnetic torque appear especially in local responses. By these reasons, I do not recommend it for publication in Nature Communications. I give several further comments below.

(1) There is no comments on the previous relevant works,

- M. Naka et al., Nat. Commun. 10, 4305 (2019).
- L. Šmejkal et al., Phys. Rev. X 12, 031042 (2022).
- S. Hayami et al., Phys. Rev. B 102, 144441 (2020).
- A. Hellenes et al., arXiv: 2309.01607.

(2) In p.3 right column, S^{even} and S^{odd} appear but there is no clear definitions of them.

(3) In Kubo formula, (1) and (2), T-even and T-odd components are often called the intra- or inter-band contributions, respectively, and it is well known that the latter is dissipationless and it is important in insulators. It is useful for some community, the familiar terminology is useful.

(4) In Eqs. (1) and (2), the spin operator is used, and it is always a local operator, and only the matrix element becomes itinerant or local depending on the basis they used. However, there is a description "To calculate the local Edelstein effect on a given sublattice, a local spin operator is used instead", which is unclear.

(5) Although there is a description "replacing the torque operator with the spin operator", no clear definition of the torque operator is given.

(6) In Fig. 1(b), it is unclear the anti-symmetric nature of the band structure, and Gamma-K-M-Gamma lines are inappropriate to show anti-symmetric behaviors. Moreover, the ordering vectors (3Q) are not given explicitly.

(7) In heterostructures, local Edelstein effect could appear due to the symmetry lowering. However, the magnetic structure would also be affected by the same symmetry lowering. How do you describe the magnetic structure in the surface/interface? Have you determined self-consistently for magnetic structures in mean-field level?

Reviewer #2:

Remarks to the Author:

This work presents a study of the Edelstein effect and current-induced torque originated due to the

non-collinear magnetic order without the need of relativistic spin orbit coupling. The authors demonstrated the existence of such nonrelativistic effects through symmetry analysis and theoretical calculations of the selected noncollinear antiferromagnets with broken local and global inversion symmetry. They also explained how these effects can be induced in centrosymmetric antiferromagnets by creating heterostructures. This work will hugely benefit to the antiferromagnetic spintronics community, and I would be happy to recommend it for publication after the authors fully address my concerns. My detailed comments and suggestions are as follows.

- 1) All the selected noncollinear antiferromagnets (e.g., Mn₃Sn (001), LuFeO₃(001), Mn₃Ir (111)) have noncollinear arrangement with allowed out-of-plane magnetic moment by symmetry. The main reason for the appearance of finite non-relativistic Edelstein effect is broken inversion symmetry either globally or locally. However, it will be inclusive if one considers antiferromagnet like Mn₃GaN where weak magnetic moment out-of-plane is not allowed by symmetry.
- 2) There are results with comparable relativistic and nonrelativistic Edelstein effects in Fig. 2 and 4. It seems to me that the manuscripts fail to explain clearly why these results are comparable.
- 3) The authors mentioned in the supplementary for Mn₃Sn calculations that the Edelstein effect for the inversion pair sublattices must be opposite. While in the main text, they claim that the Mn₃Sn/non-magnetic heterostructures might also have non-relativistic effects. In such cases, what will be the effect of the local Edelstein effects in the even layered Mn₃Sn and odd layered Mn₃Sn? In addition, can the author explain the contribution of the induced Mn₃Sn/non-magnetic heterostructures like Mn₃Ir?
- 4) The manuscript shows that the local T-odd Edelstein effect in LuFeO₃ is finite within the band gap. There is no clear explanation why it is happening and is this property specific to the LuFeO₃ or globally noncentrosymmetric antiferromagnet. Can this phenomenon also exist in the locally noncentrosymmetric antiferromagnet like Mn₃Sn?
- 5) What are the reasons behind the choice of the Γ parameters value? How does the observed value differ with the choice of the Γ operators within the band gap and other case considered in the manuscript?

Minor Issues

- 1) Figures labeling has randomly patterned directions on going from a to b to c toFigure 4 is not labeled at all and there are figures with overlap of the numbers on the visual images. It would be great if the authors can work on these so that readers can follow the manuscript more easily.
- 2) Figure 3 has typo errors in the captions.
- 3) The authors have listed allowed Edelstein effect tensors in the supplementary. It will be clearer for the reader if the author illustrates symmetry analysis for the allowed tensor components at least in the selected material.

RESPONSE TO REVIEWERS' COMMENTS

Reviewer #1 (Remarks to the Author):

The authors have discussed the torque and Edelstein effects emerged in noncollinear magnets where the relativistic spin-orbit coupling plays a negligible role. They first discussed symmetry conditions for the presence of spin splitting in particular noncolinear magnetic structure with 3 components, and revealed that the directions of S^{even} and S^{odd} are classified by symmetry operations. Then, they demonstrate the spin splitting and Edelstein effect for some representative materials. They also found that in some class of compounds, the non-relativistic origin of magnetic torque exists in insulators. The contents are worth publishing in some form, but the impact of the manuscript is unclear. The potential weaknesses are: (i) it is hard to extract the authors' new assertions as compared with the previous relevant works, and (ii) the authors seem to estimate the responses quantitatively for realistic heterostructure, but the model parameters are chosen as arbitrary, which means that quantitative conclusions are valid or not in general sense and it is not clear which conclusion is applied widely to similar compounds in general. By using the heterostructures having lower symmetry, it is of course expected that the non-relativistic origin of spin splitting and consequent Edelstein and magnetic torque appear especially in local responses. By these reasons, I do not recommend it for publication in Nature Communications. I give several further comments below.

(i) Position of our manuscript compared to previous works:

We understand that the referee's first concern is about the novelty of our work compared to previous predictions. In a nutshell, **our manuscript studies a non-relativistic self-induced torque, whereas the papers cited by the referee address a completely separate phenomenon, the (antisymmetric) spin splittings.** More specifically:

- All the papers cited by the referee focus on the momentum-symmetric or antisymmetric spin-splitting in the Brillouin zone, which can be addressed experimentally with ARPES for instance. In contrast, we investigate the emergence of current-driven *torque* that can be measured experimentally using various conventional setups and potentially harvested for applications including memory or high-frequency oscillators in the broader field of spintronics, as recently demonstrated using spin-orbit torque [Tsai et al., Nature 580, 608 (2020); Higo et al., Nature 607, 474 (2022)].
- Our investigation of the spin-orbit-free Edelstein effect is not limited to non-centrosymmetric antiferromagnets with momentum-antisymmetric spin-splitting but covers a much broader class of systems including heterostructures and centrosymmetric antiferromagnets with local torques, which to the best of our knowledge has never been addressed before.
- Importantly, our study demonstrates that the momentum-antisymmetric spin-splitting is *not sufficient or even necessary* to produce the torque and that the symmetry analysis should be extended to the sublattice level. In this respect, the group theory analysis provided in the manuscript is radically different from that of Hayami 2020. Hayami's analysis focuses on momentum-antisymmetric spin-splitting

whereas ours concerns the existence of both local and non-local torques, and does not necessarily rely on the spin-splitting.

We provide a detailed discussion of the four references brought by the reviewer.

M. Naka et al., Nat. Commun. 10, 4305 (2019).

We thank the referee for bringing this paper to our attention. In this paper, the authors propose that *spin current* may be generated in a class of organic antiferromagnets even in the absence of spin-orbit coupling. This work differs from ours by several aspects: (i) the antiferromagnetic arrangement is collinear whereas ours is non-collinear, (ii) the system is inversion symmetric whereas ours is inversion symmetry-broken, which results in (iii) momentum-symmetric spin-splitting of the band structure whereas ours is momentum-antisymmetric. Finally, (iv) the authors focus on the generation of a pure *spin current* whereas we focus on the torque emerging from non-equilibrium *spin density*. We notice that the existence of a spin current emerging in noncollinear antiferromagnets without spin-orbit coupling was proposed by one of us a few years ago, Zelezny et al., Physical Review Letters 119, 187204 (2017).

Naka et al.'s proposal falls into the domain of "spin-split antiferromagnets", also called "altermagnets", where spin-current generation is due to the anisotropic orbital configuration of the antiferromagnetic sublattices. This situation was simultaneously reported by Hayami et al. Journal of the Physical Society of Japan 88, 123702 (2019), Yuan et al. PRB 102, 014422 (2020), and Smejkal et al. Science Advances 6, eaaz8809 (2020). In L. Šmejkal et al., Phys. Rev. X 12, 031042 (2022) and Šmejkal et al. PRX 12, 040501 (2022), this class of antiferromagnets, whose magnetic sublattices are related by rotation symmetry, has been tagged "altermagnets".

In all these studies, the antiferromagnet has a collinear arrangement (but non-collinear altermagnets also exist in principle) and the spin current is polarized along the Néel order. Such a spin-current cannot exert a spin torque in the volume of the antiferromagnet, but only at the interfaces, as reported by some of us [Ghosh et al. Physical Review Letters 128 (9), 097702 (2022)]. **Our situation is therefore markedly different from these references.**

We thank the referee for bringing S. Hayami et al., Phys. Rev. B 102, 144441 (2020) and A. Hellenes et al., arXiv: 2309.01607 to our attention. We were aware of S. Hayami et al. Physical Review B 101, 220403 (2020), but not of his follow-up work on this matter. These three references focus on the momentum-antisymmetric spin-splitting, that is – in certain cases - a condition for the emergence of the Edelstein effect. Hayami et al., Phys. Rev. B 102, 144441 (2020) in particular provides a remarkably detailed predictive analysis of the band structure of spin-orbit-free antiferromagnets based on an augmented multipole basis, covering both centrosymmetric and non-centrosymmetric situations. As explained above, our own analysis takes a different perspective and does not focus on the spin-splitting but rather on the current-driven torque.

In order to clarify the position of our manuscript compared to these previous studies and emphasize its singularity, we have added a comment in the introduction.

(ii) Validity of the generality of our predictions:

The objective of the manuscript is to demonstrate that nonrelativistic local torques are possible and, most importantly, **sizable**. For this purpose, two of the calculations reported in the manuscript are performed using *ab initio* techniques, whereas the two others use model Hamiltonians.

The 3Q model discussed at the beginning of the manuscript is not meant to be predictive but rather pedagogical. Considering the complexity of the physics at stake, we chose to include such a discussion in order to reach a broader audience, beyond the handful of experts on antiferromagnetic spintronics. That being said, the parameters of this model are very standard, with the hopping and exchange energies being of the order of 1 eV, as is usually the case in magnetic metals. For instance, the groundbreaking prediction of the anomalous Hall effect in noncollinear antiferromagnets by Chen et al., Physical Review Letters 112, 017205 (2014), used a similar pedagogical model to introduce the concept, using comparable parameters.

The case of the nonrelativistic torque in “non-centrosymmetric heterostructures” follows a dual objective. First, we aim to demonstrate that interfacial inversion symmetry breaking is sufficient to enable current-driven torque in noncollinear antiferromagnets. From the standpoint of antiferromagnetic spintronics, this demonstration is crucial since such heterostructures can be fabricated and engineered using sputtering and epitaxial deposition techniques available in most spintronics labs. It offers much more flexibility for optimization than the bulk materials studied in the main text or listed in the supplemental materials. A second objective was, of course, to show that the interfacial inversion symmetry breaking was sufficient to induce an experimentally observable torque.

In the course of our investigation, we did compute the torque in a realistic $\text{Mn}_3\text{Sn}/\text{Ru}(0001)$ heterostructure computed from first principles. We point out that performing *ab initio* transport calculations in this system is a very difficult problem due to the large number of atoms in the unit cell. It is therefore out of the scope of the present work. Nonetheless, we hereby provide preliminary results for the sake of the discussion. In this initial work, we were able to compute the torque in Mn_3Sn [4 ML]/ $\text{Ru}(0001)$ [6 ML] (ML=monolayer), with one specific noncollinear magnetic configuration, which is different from the magnetic structure of bulk Mn_3Sn . Such a very small structure, which is in itself quite complex to compute *ab initio*, possesses more than 50 atoms in the supercell, which renders the Wannier projection prohibitive.

Nonetheless, these preliminary calculations demonstrate that the nonrelativistic torque is of comparable magnitude as the ones computed in the other systems. We therefore decided to represent the heterostructure using a model Hamiltonian with standard hopping and exchange energies (typically 1 eV), as generally performed in the literature [e.g., Chen et al., Physical Review Letters 112, 017205 (2014); Mook et al. Physical Review Research 2, 023065 (2020)]. Remarkably, the magnitude of the torque we obtain in this model heterostructure is comparable to the one obtained using our *ab initio* calculations.

For the sake of transparency, we provide below the summary of our *ab initio* calculations.

These calculations are now inserted in the Supplemental Material for completeness.

The present calculations are performed in the framework of the density functional theory (DFT) using VASP code. The parameter conditions mirror those outlined in the main text for bulk calculations of Mn_3Sn and LuFeO_3 . The Brillouin zone integration is executed employing an $(11 \times 11 \times 1)$ k-points mesh. The interface model was calculated using a $\text{Mn}_3\text{Sn}/\text{Ru}$ surface, which represents a slab with 4 Mn_3Sn and 6 Ru layers, stacked in the hexagonal growth direction $[0001]$. The slabs are separated from their periodic replicas in the $[0001]$ direction by a 17 \AA vacuum layer. During the geometry optimization stage, relaxation of all slab atoms is performed until the forces acting on them do not exceed 0.01 eV/\AA . All calculations were carried out without considering the spin-orbit coupling interaction. In the Wannierization process, we employed the d orbitals for the Ru and Mn atoms, while the p orbitals were utilized for the Sn atoms. The frozen energy window was set to $E_{\text{Fermi}} = +1 \text{ eV}$. In our linear response calculations, we employed the Linres code with a $480 \times 480 \times 1$ k-mesh.

Figure R1: T-odd components of the local Edelstein effect projected on the three magnetic sublattices, as indicated on the crystal structure.

Figure R2: T-even components of the local Edelstein effect projected on the three magnetic sublattices, as indicated on the crystal structure.

The results for the T-odd and T-even components of the torque, projected on the three magnetic sublattices as indicated in the sketch of the heterostructure are reported in Figs. R1 and R2, respectively. Two comments are in order: first, the overall magnitude of the effect is comparable to that reported in Fig. 5 of the main text and computed with a model system. From our viewpoint, this is not surprising as the band structure of this metallic heterostructure is very dense and the spin transport is rather governed by the interplay between the magnetic configuration, the bandwidth, and the interfacial orbital hybridization. In the presence of a large number of orbitals, as is the case here, one expects that the details in the band structure should have a reduced impact.

A second point we wish to make is that this highly computationally demanding heterostructure only possesses four magnetic planes and is not adapted to study the spin propagation inside the antiferromagnet itself. The model system reported in the main text is computationally lighter, allowing for the modeling of a much thicker antiferromagnetic layer and hence the description of the torque profile.

(2) In p.3 right column, S^{even} and S^{odd} appear but there is no clear definitions of them.

We apologize for the lack of clarity; it has been corrected.

(3) In Kubo formula, (1) and (2), T-even and T-odd components are often called the intra- or inter-band contributions, respectively, and it is well known that the latter is dissipationless and it is important in insulators. It is useful for some community, the familiar terminology is useful.

We have added a note below Eqs. (1) and (2).

(4) In Eqs. (1) and (2), the spin operator is used, and it is always a local operator, and only the matrix element becomes itinerant or local depending on the basis they used. However, there is a description "To calculate the local Edelstein effect on a given sublattice, a local spin operator is used instead", which is unclear.

Here what we mean by "local spin operator" is the projection of the spin operator on the given sublattice. We have clarified this in the manuscript.

(5) Although there is a description "replacing the torque operator with the spin operator", no clear definition of the torque operator is given.

We do not compute the spin torque in the manuscript but rather focus on the spin density that gives rise to such a torque. It is therefore not necessary to define the torque operator and to avoid confusion, we removed this part of the sentence.

(6) In Fig. 1(b), it is unclear the anti-symmetric nature of the band structure, and Gamma-K-M-Gamma lines are inappropriate to show anti-symmetric behaviors. Moreover, the ordering vectors (3Q) are not given explicitly.

The referee is correct. In this figure, our intention was not to show the antisymmetric spin-splitting in the band structure itself, but rather on the Fermi surface. By the way, we would like to draw the referee's attention to the fact that the spin texture is not purely antisymmetric; it rather shows a dissymmetry that is at the origin of the spin torque, as illustrated in Fig. R3.

To acknowledge the referee's comment, we have replaced Fig. 1(f) with Fig. R3(c) and reorganized the labels for better readability.

Figure R3: Spin-projected band structure for (a) S_x , (b) S_y and (c) S_z .

(7) In heterostructures, local Edelstein effect could appear due to the symmetry lowering. However, the magnetic structure would also be affected by the same symmetry lowering. How do you describe the magnetic structure in the surface/interface ? Have you determined self-consistently for magnetic structures in mean-field level ?

The referee is in principle correct. If the substrate were to exhibit large spin-orbit coupling, the Dzyaloshinskii-Moriya, and interfacial magnetic anisotropy interactions would arise and possibly impact the magnetic ordering. In this case, self-consistent determination of the magnetic arrangement would require an extensive density functional theory investigation. This would be particularly relevant in the case where spin-orbit coupling is strong, e.g., with substrates made of 5d materials such as Pt, Ta, or W. In addition, the interfacial magnetic configuration would be very dependent on the given heterostructure, thickness, and growth conditions.

Performing a full computational study combining the optimization of the magnetic texture together with transport calculations is by itself a highly challenging task. In fact, in all the first principles calculations of spin-orbit torque studies we are aware of (e.g., Freimuth et al., Physical Review B 90, 174423 (2014); Belashchenko et al., Physical Review Materials 3, 011401(R) (2019)), it is considered that these additional interactions do not significantly impact the magnetic texture. Qualitatively, the argument commonly adopted is that for a sufficiently thick layer, the magnetic order is expected to be that of the bulk.

In the present work, we do not aim to give quantitative predictions for specific heterostructures and such simulations are well beyond the scope of our paper.

Reviewer #2 (Remarks to the Author):

This work presents a study of the Edelstein effect and current-induced torque originated due to the non-collinear magnetic order without the need of relativistic spin orbit coupling. The authors demonstrated the existence of such nonrelativistic effects through symmetry analysis and theoretical calculations of the selected noncollinear antiferromagnets with broken local and global inversion symmetry. They also explained how these effects can be induced in centrosymmetric antiferromagnets by creating heterostructures. This work will hugely benefit to the antiferromagnetic spintronics community, and I would be happy to recommend it for publication after the authors fully address my concerns. My detailed comments and suggestions are as follows.

We thank the reviewer for his/her support and constructive remarks.

1) All the selected noncollinear antiferromagnets (e.g., Mn₃Sn (001), LuFeO₃(001), Mn₃Ir (111)) have noncollinear arrangement with allowed out-of-plane magnetic moment by symmetry. The main reason for the appearance of finite non-relativistic Edelstein effect is broken inversion symmetry either globally or locally. However, it will be inclusive if one considers antiferromagnet like Mn₃GaN where weak magnetic moment out-of-plane is not allowed by symmetry.

We thank the reviewer for this comment. In all the noncollinear antiferromagnets mentioned, the net moment is of relativistic origin, i.e., the canting requires spin-orbit coupling. Since our focus is on the non-relativistic torque, effects related to spin-orbit coupling only come as a correction. We therefore do not believe that accounting for spin canting has any sizable nor qualitative influence on our results. Concerning Mn₃GaN, it is cubic and possesses (up to a rotation) the same magnetic configuration Mn₃Ir, and therefore the local torque vanishes by symmetry. We added a comment in the main text to clarify this point.

2) There are results with comparable relativistic and nonrelativistic Edelstein effects in Fig. 2 and 4. It seems to me that the manuscripts fail to explain clearly why these results are comparable.

The reason why the torque computed with (dashed lines) and without spin-orbit coupling (solid lines) are comparable is because spin-orbit coupling is weak. As a matter of fact, close to the gap, the spin-orbit coupling remains much smaller than the exchange, and relativistic corrections are negligible compared to the bare nonrelativistic Edelstein effect. In the manuscript, we commented on this aspect: *"Including the spin-orbit coupling does not change the results substantially, similar to previous calculations of the spin Hall effect in this material."*

3) The authors mentioned in the supplementary for Mn₃Sn calculations that the Edelstein effect for the inversion pair sublattices must be opposite. While in the main text, they claim that the Mn₃Sn/non-magnetic heterostructures might also have non-relativistic effects. In such cases, what will be the effect of the local Edelstein effects in the even layered Mn₃Sn and odd layered Mn₃Sn? In addition, can the author explain the contribution of the induced Mn₃Sn/non-magnetic heterostructures like Mn₃Ir?

We have included the results of DFT-based calculations of Mn₃Sn/Ru heterostructure in the Supplemental Materials (see our detailed response to Reviewer 1's first comment). However, we note that these calculations are preliminary and use a different magnetic structure than the one we have considered for the bulk.

We have also included tight-binding calculations of Mn₃Sn/Ru heterostructure analogous to the calculations we have performed for the Mn₃Ir heterostructure. These calculations clearly demonstrate the breaking of inversion symmetry and we then expect that the torque can efficiently manipulate the magnetic order. We do not expect any significant dependence on whether the number of layers is even or odd since the inversion symmetry remains broken in either case. In the case of a free-standing layer or weak coupling to the non-magnetic layer, there could be a dependence on whether the number of layers is even or odd, however, studying such an effect would require investigating the magnetic dynamics of the structure. This is an interesting topic for further study but it lies beyond the scope of the present work.

4) The manuscript shows that the local T-odd Edelstein effect in LuFeO₃ is finite within the band gap. There is no clear explanation why it is happening and is this property specific to the LuFeO₃ or globally noncentrosymmetric antiferromagnet. Can this phenomenon also exist in the locally noncentrosymmetric antiferromagnet like Mn₃Sn?

We thank the reviewer for this important comment. The T-odd Edelstein effect is related to the Berry curvature in spin-momentum space and, as such, it comes from matrix elements between occupied and unoccupied states. In other words, it is a Fermi-sea contribution, as studied in the context of spin-orbit torque for instance [Garate Physical Review Letters 104, 146802 (2010); Kurebayashi et al., Nature Nanotechnology 9, 211 (2014); Li et al., Physical Review B 91, 134402 (2015) etc.]. Because this term is associated with interband transitions, it does not necessarily vanish in the gap. An illustration of this peculiarity can be found in Garate Physical Review Letters 104, 146802 (2010) in the case of a magnetic topological insulator; in this paper, the authors attribute this effect to topology. Yet, the nonvanishing T-odd Edelstein effect in the insulating regime has not been thoroughly studied, and, based on our results, we believe that it is not restricted to topological systems.

Notice that the T-odd Edelstein effect reported on LuFeO₃ is projected on the sublattices. Therefore, this effect should exist in both globally and locally non-centrosymmetric antiferromagnets. In the present manuscript, we have been focusing on metallic cases, but the suggestion raised by the reviewer opens interesting future directions.

We added a comment in the main text to clarify the physical origin of the nonvanishing torque in the insulating regime.

5) What are the reasons behind the choice of the Γ parameters value? How does the observed value differ with the choice of the Γ operators within the band gap and other case considered in the manuscript?

The Γ parameter is usually chosen to match the conductivity computed numerically with the experimental value. For consistency, we kept $\Gamma=0.01$ eV all along the manuscript. Based on the linear response theory, the T-even Edelstein effect is inversely proportional to Γ , whereas the T-odd effect is independent of Γ in the limit of weak disorder. We added a comment to the manuscript.

Minor Issues

1) Figures labeling has randomly patterned directions on going from a to b to c toFigure 4 is not labeled at all and there are figures with overlap of the numbers on the visual images. It would be great if the authors can work on these so that readers can follow the manuscript more easily.

We have updated the figures as requested.

2) Figure 3 has typo errors in the captions.

The caption has been corrected.

3) The authors have listed allowed Edelstein effect tensors in the supplementary. It will be clearer for the reader if the author illustrates symmetry analysis for the allowed tensor components at least in the selected material.

We have added a section in the Supplementary Material describing the procedure for the symmetry analysis. In practice, we always do the symmetry analysis using our own open-source code Symmetr, and have also added a brief description of how the code can be used to obtain the response tensors.

Reviewers' Comments:

Reviewer #1:

Remarks to the Author:

The authors have addressed most of the points I raised and have added clarifications to the text and supplementary material that make the paper more understandable. I consider that the current improved version is suitable for publication.

Reviewer #2:

Remarks to the Author:

The authors have responded well to the questions raised in the review. Moreover, they have added extra calculations on Mn₃Sn/Ru heterostructure. Inclusion of the new calculations and symmetry analysis informations have added clarity to the origin of Edelstein effect due to non collinear arrangement of magnetic moment in weakly spin orbit coupled antiferromagnetic system. This work will largely benefit antiferromagnetic spintronics community. As my concerns have been answered well, I am happy to recommend for the publications.